# Relationship between MODIS Derived NDVI and Yield of Cereals for Selected European Countries

Ewa Panek * and Dariusz Gozdowski 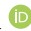

Department of Biometry, Warsaw University of Life Sciences, Nowoursynowska 159, 02-776 Warsaw, Poland; dariusz_gozdowski@sggw.edu.pl
* Correspondence: ewa_panek@sggw.edu.pl

**Abstract:** In this study, the relationships between normalized difference vegetation index (NDVI) obtained based on MODIS satellite data and grain yield of all cereals, wheat and barley at a country level were analyzed. The analysis was performed by using data from 2010–2018 for 20 European countries, where percentage of cereals is high (at least 35% of the arable land). The analysis was performed for each country separately and for all of the collected data together. The relationships between NDVI and cumulative NDVI (cNDVI) were analyzed by using linear regression. Relationships between NDVI in early spring and grain yield of cereals were very strong for Croatia, Czechia, Germany, Hungary, Latvia, Lithuania, Poland and Slovakia. This means that the yield prediction for these countries can be as far back as 4 months before the harvest. The increase of NDVI in early spring was related to the increase of grain yield by about 0.5–1.6 t/ha. The cumulative of averaged NDVI gives more stable prediction of grain yield per season. For France and Belgium, the relationships between NDVI and grain yield were very weak.

**Keywords:** normalized difference vegetation index—NDVI; grain yield; cereals; prediction

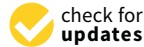



## 1. Introduction

The most important crops in European Union (EU 28) agriculture are cereals because 53% of the arable land is dedicated to the harvested area for all cereal species [1]. The most important crop among cereals is wheat which covers around 25% of total arable area of EU. These crops have very high importance for food protection because of the high variability of grain yields of cereals from one year to another. The forecast of the grain yields for certain season is important [2] and should be prepared as early as possible because it allows for proper trading and planning of grain stocks by farmers and trading companies [3,4]. The yield forecasts can be prepared at various spatial levels, from forecasts for individual crop fields to regional forecasts for all countries or even on a worldwide scale [5]. Grain yield forecasts are based on various methods, including statistical models where meteorological data and historical yields are used for prediction [2–5]. Remote sensing data from satellites more often are used as input data in crop models for grain yield forecasts. AVHRR and MODIS are the most popular satellite sensors to obtain within-season information to forecast final yield at regional scale [6–8], while for smaller areas, the most commonly used data are derived from Landsat and Sentinel satellites [9–11]. Multispectral data in the range of visible light and near-infrared are used for calculation of vegetation indices which are used as predictors of cereal grain yield. The most common vegetation index used for such purpose is normalized difference vegetation index (NDVI) which is based on reflectance in red and near-infrared light spectrum [12]. In the simplest approach, the mid-season NDVI is treated as a cereal grain yield predictor where simple linear regression is applied for yield forecasts [13,14]. Thanks to the use of such methods, it was possible, in some studies, to forecast grain yield 2–3 months before the harvest. More advanced regression models for yield forecast use time series of NDVI which allow one to obtain better prediction. Multiple regression, where predictors are NDVI values

from different growth stages of the crop, is one of the methods which can be applied [15,16]. Cumulative NDVI (cNDVI) is based on time series until pre-heading stage or later stages can be used as a predictor of cereal grain yield [17–20]. Yield prediction can be performed using vegetation condition index (VCI), which is expressed in the percentage and gives an idea of where the observed value is situated between the extreme values (minimum and maximum) during the previous years [21]. The approach where VCI is yield predictor of wheat yield for European countries allows one to obtain good yield prediction by using partial least square regression [22]. Not only NDVI is used for yield forecasting. Other vegetation indices such as SR (Simple Ratio Index), RVI (Ratio Vegetation Index), EVI (Enhanced Vegetation Index), NDRE (Normalized Difference Red Edge Index), FA-PAR (Fraction of Absorbed Photosynthetically Active Radiation), SAVI (Soil-adjusted vegetation index, GNDVI (Green Normalized Difference Vegetation Index) are also used [23–28]. Yield forecasting at regional level, using satellite remote sensing, can be performed by using mean values of the vegetation indices for the total area or only for the total cropland or for the crop masks for individual crops [13,18,29,30].

The aim of this study is to evaluate the forecast accuracy of grain yield of cereal crops for selected European countries based on average and cumulative MODIS derived NDVI for different dates in the 2010–2018 years. This is important because forecasting yields and monitoring the state of agricultural production are key to maintaining Europe's food security. This article is intended to show not only how the yield is related to NDVI changes in individual European countries, but also to indicate in which countries, based on NDVI, it is not possible to accurately predict yields of cereals.

## 2. Materials and Methods

### 2.1. Grain Yield Data

Twenty European countries were selected for this study (Table 1 and Figure 1). The main criterion for the selection of the country was its high percentage of cereals in arable land (at least 35%). Moreover, we considered the predominance of crops like wheat and barley. Most of the countries are located in central and western Europe, but most of the Mediterranean countries were excluded because of the low percentage of wheat and barley.

**Table 1.** The countries selected for this study and the area of cereals as percentage of the arable land in 2018 (based on FAOSTAT database [1]).

| Country | All Cereals | Wheat | Barley |
|---|---|---|---|
| Austria | 58.8% | 22.0% | 10.5% |
| Belgium | 36.4% | 23.4% | 5.0% |
| Bulgaria | 52.1% | 34.7% | 3.0% |
| Croatia | 56.9% | 16.9% | 6.2% |
| Czechia | 53.7% | 32.8% | 13.0% |
| Denmark | 59.8% | 18.0% | 33.6% |
| Estonia | 51.2% | 22.6% | 20.2% |
| Finland | 40.5% | 7.9% | 18.1% |
| France | 49.4% | 28.3% | 9.6% |
| Germany | 51.8% | 25.8% | 13.8% |
| Hungary | 55.0% | 23.8% | 5.7% |
| Ireland | 59.4% | 13.2% | 42.0% |
| Latvia | 52.7% | 32.3% | 9.2% |
| Lithuania | 59.8% | 36.7% | 10.7% |
| Poland | 71.6% | 22.2% | 8.9% |
| Romania | 61.5% | 24.7% | 4.9% |
| Slovakia | 55.3% | 30.0% | 9.2% |
| Slovenia | 54.1% | 15.2% | 11.4% |
| Sweden | 35.9% | 14.5% | 14.1% |
| United Kingdom | 51.1% | 28.7% | 18.7% |

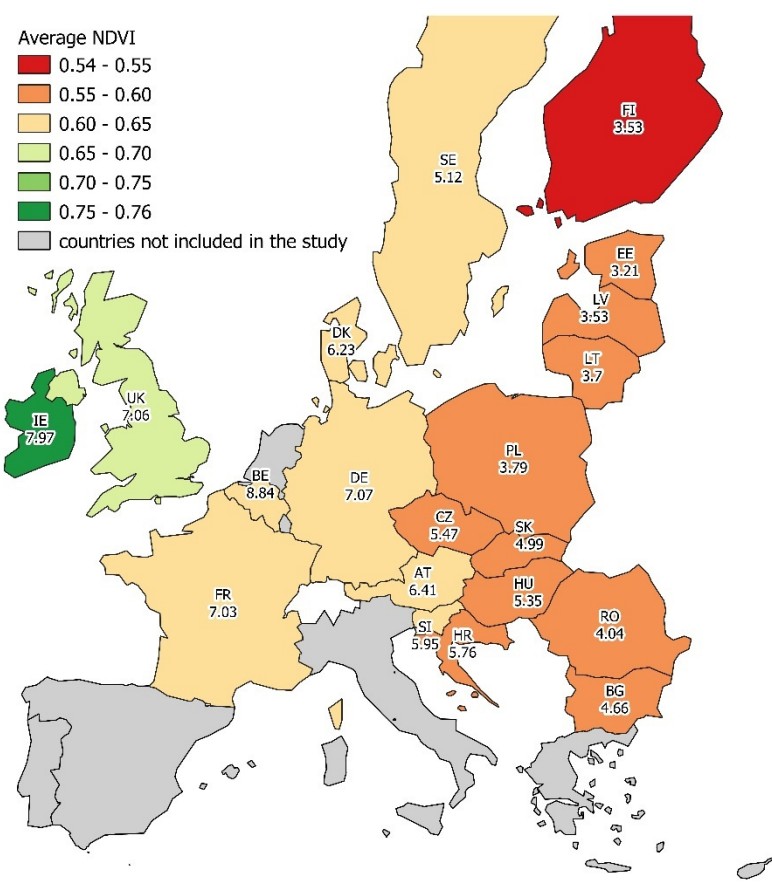

**Figure 1.** The map of the countries selected for the study presenting average normalized difference vegetation index (NDVI) (in different colors) for the period from 26 February to 12 August in 2010–2018 and average grain yield (values in labels).

The grain yield data and area of the arable land were obtained from FAOSTAT database [1]. Mean value of the percentage of all cereal crops in the arable land was 53.3% (from 35.9% in Sweden to 71.6% in Poland), where FAOSTAT defines cereals as: wheat, rice paddy, barley, maize, popcorn, rye, oats, millets, sorghum, buckwheat, quinoa, fonio, triticale, canary seed, mixed grain, cereals nes. However, in most of the studied countries wheat (especially winter wheat) and barley have a large share, as can be seen in Table 1. Other crop species have small or negligible shares. Wheat covered an average of 23.7% (from 7.9% in Finland to 36.7% in Lithuania) of arable land, while barley area was in average 13.4% (from 3.0% in Bulgaria to 42.0% in Ireland). Moreover, the typical sowing dates and dates of main phenological stages of winter wheat are presented in Table S1. The differences between countries for winter crops in early spring are not very large because typical growth stage in early spring for winter cereals such as wheat or barley is the tillering stage. Bigger differences occur in later growth stages, e.g., flowering stage which can be different by about one month between the studied countries; even within the country, the time differences are up to 3 weeks.

### 2.2. Remote Sensing Data

Time series NDVI averaged for countries considered in this study was downloaded from GIMMS Global Agricultural Monitoring (GLAM) system (https://glam1.gsfc.nasa.gov/ (accessed on 12 January 2021)) hosted by USDA and NASA [31]. The GLAM system provides 8-day composited NDVI datasets based on satellite imagery acquired by MODIS sensors from Terra satellite platform. These datasets are derived from MOD09 product (MODIS collection 6) at spatial resolution 250 or 500 m. The first 8-day period included in this analysis was from 26 February to 3 March and the last 8-day period was from 5 August

to 12 August. The same periods were for all 9 years which were analyzed, i.e., from 2010 to 2018 and for each of the countries. The NDVI values were averaged for croplands using crop mask GFSAD30 2015 Crops, which is crop mask at spatial resolution 30 m developed by NASA project Global Food Security-Support Analysis Data (https://croplands.org (accessed on 12 January 2021)) [32]. Cumulative NDVI (cNDVI) was calculated for the end of each 8-day period as a product of mean NDVI for each period and number of days.

$$\text{cNDVI} = \sum_{i=1}^{k} (\text{NDVI}_i \times 8) \tag{1}$$

where i is i-th 8-day period and k is the last period for which cNDVI is calculated.

The values of NDVI for northern European countries were filled or replaced by average NDVI from neighboring periods or by the NDVI value from the nearest next period, because for some 8-day periods there was lack of data or the data were based on small number of pixels (less than 1000) for beginning of March.

### 2.3. Statistical Analysis

The relationship between NDVI for individual 8-day periods and cNDVI was calculated across years separately for each country. The correlation coefficients and linear regression were applied for evaluation of the relationship. For each regression model the coefficients of determination ($R^2$) and standard errors of predictions were calculated. The dates for the best yield prediction based on NDVI and cNDVI were selected for each country.

## 3. Results

### 3.1. Spatiotemporal Variability of Grain Yield and NDVI

The highest averaged grain yield of all cereals for total period of the study (2010–2018) was observed in Belgium (8.84 t/ha) (Table 2). The grain yield in Belgium was very high for barley and wheat (8.09 and 8.66 t/ha respectively for these crops). The second highest grain yield of cereals for that period was observed in Ireland (7.97 t/ha), which is where the highest grain yield of wheat was observed (9.27 t/ha). High yields (about 7.0 t/ha) but slightly lower were observed in the United Kingdom, Germany and France. The smallest average grain yield was observed for the period of this study (2010–2018) in Estonia (3.21 t/ha). The small yield was observed as well for Latvia and Finland (mean 3.52 t/ha for both countries), Lithuania (3.70 t/ha) and Poland (3.79 t/ha). Temporal (between years) variability of grain yields expressed as standard deviation was usually higher in countries where higher yields were observed. Value of ratio of the standard deviation to mean grain yield (coefficient of variation—CV) was the highest for Romania (26%) and Estonia (21%) which indicates high temporal variability of grain yield across years. Much smaller temporal variability (CV in range 7–9%), which indicates relatively small differences between grain yields in years was observed for the following countries: Austria, Finland, France, Germany, Poland and the United Kingdom. In most of the countries, the grain yield of wheat was much higher (on average up to 34% for Latvia) in comparison to grain yield of barley. The only exception was Austria where both averaged grain yields (barley and wheat) were almost the same (5.31 t/ha).

**Table 2.** Means ± standard deviations (and coefficients of variations—CV) for grain and NDVI for all years included in the study (2010–2018).

| Area | Cereals. Total Grain Yield (t/ha) | Wheat Grain Yield (t/ha) | Barley Grain Yield (t/ha) | Average NDVI [1] (CV in % within the Seasons and between Years) | Final Cumulative NDVI—cNDVI [2] (CV in % Across Years) |
|---|---|---|---|---|---|
| Austria | 6.41 ± 0.55 | 5.31 ± 0.69 | 5.31 ± 0.57 | 0.61 ± 0.11 (18.2) | 102.4 ± 3.0 (2.9) |
| Belgium | 8.84 ± 0.87 | 8.66 ± 0.88 | 8.09 ± 0.87 | 0.65 ± 0.07 (11.2) | 109.2 ± 3.2 (3.0) |
| Bulgaria | 4.66 ± 0.62 | 4.36 ± 0.57 | 3.97 ± 0.41 | 0.58 ± 0.10 (16.6) | 98.0 ± 5.3 (5.4) |
| Croatia | 5.76 ± 0.75 | 5.10 ± 0.63 | 4.16 ± 0.50 | 0.60 ± 0.10 (16.9) | 101.5 ± 3.8 (3.8) |
| Czechia | 5.47 ± 0.62 | 5.68 ± 0.73 | 4.96 ± 0.58 | 0.60 ± 0.13 (21.7) | 101.1 ± 4.0 (4.0) |
| Denmark | 6.23 ± 0.64 | 7.23 ± 0.70 | 5.53 ± 0.54 | 0.62 ± 0.14 (22.1) | 103.5 ± 6.6 (6.4) |
| Estonia | 3.21 ± 0.69 | 3.49 ± 0.75 | 3.17 ± 0.71 | 0.58 ± 0.19 (32.6) | 96.6 ± 3.4 (3.5) |
| Finland | 3.53 ± 0.28 | 3.77 ± 0.41 | 3.59 ± 0.26 | 0.54 ± 0.20 (36.5) | 90.4 ± 2.8 (3.1) |
| France | 7.03 ± 0.57 | 6.95 ± 0.71 | 6.32 ± 0.51 | 0.64 ± 0.06 (9.1) | 108.2 ± 2.2 (2.1) |
| Germany | 7.07 ± 0.57 | 7.58 ± 0.60 | 6.49 ± 0.63 | 0.63 ± 0.10 (15.4) | 106.6 ± 4.1 (3.8) |
| Hungary | 5.35 ± 0.86 | 4.68 ± 0.66 | 4.34 ± 0.67 | 0.57 ± 0.10 (18.2) | 95.5 ± 4.3 (4.5) |
| Ireland | 7.97 ± 0.82 | 9.27 ± 1.08 | 7.57 ± 0.77 | 0.76 ± 0.05 (7.1) | 127.5 ± 3.8 (3.0) |
| Latvia | 3.53 ± 0.62 | 3.99 ± 0.68 | 2.98 ± 0.54 | 0.59 ± 0.19 (31.4) | 99.3 ± 4.1 (4.1) |
| Lithuania | 3.70 ± 0.59 | 4.27 ± 0.68 | 3.27 ± 0.52 | 0.58 ± 0.17 (30.0) | 97.2 ± 4.0 (4.1) |
| Poland | 3.79 ± 0.32 | 4.46 ± 0.33 | 3.59 ± 0.31 | 0.59 ± 0.13 (23.0) | 98.4 ± 3.2 (3.2) |
| Romania | 4.04 ± 1.05 | 3.72 ± 0.78 | 3.38 ± 0.69 | 0.58 ± 0.11 (19.6) | 96.7 ± 5.7 (5.9) |
| Slovakia | 4.99 ± 0.90 | 4.68 ± 0.88 | 4.07 ± 0.81 | 0.58 ± 0.13 (22.5) | 97.2 ± 3.9 (4.1) |
| Slovenia | 5.95 ± 0.59 | 4.97 ± 0.38 | 4.53 ± 0.30 | 0.65 ± 0.10 (16.0) | 110.0 ± 3.0 (2.7) |
| Sweden | 5.12 ± 0.80 | 6.05 ± 0.92 | 4.57 ± 0.71 | 0.61 ± 0.14 (23.1) | 102.8 ± 4.2 (4.1) |
| United Kingdom | 7.06 ± 0.53 | 7.88 ± 0.68 | 5.95 ± 0.38 | 0.70 ± 0.08 (11.2) | 117.5 ± 4.5 (3.8) |

[1] Average NDVI was calculated for the whole period from 26th of February to 12th of August for all years (2010–2018) separately for each country; [2] Final cumulative NDVI (cNDVI) was calculated for the end of the period from 26th of February to 5th of August for all years (2010–2018) separately for each country.

The highest averaged NDVI for all years of the study (2010–2018) and period included in the study (from the end of February to beginning of August) was observed for Ireland (0.76). It was much higher in comparison to all other countries included in the study. Other countries where averaged NDVI values were high are: United Kingdom (0.70), Slovenia (0.66), Belgium (0.65), France (0.64) and Germany (0.64). By contrast, the lowest averaged NDVI was observed for Finland (0.54). Low averaged values of NDVI were observed for Bulgaria (0.58), Estonia (0.58), Hungary (0.57), Lithuania (0.58), Poland (0.59), Romania and Slovakia (0.58). For most of the countries where low values of NDVI were observed, relative temporal variability of NDVI was high (coefficient of variation up to 36.5% for Finland) while for the countries with the highest NDVI, relative temporal variability (together within the seasons and between years) of NDVI was much smaller (the lowest CV = 7.1% for Ireland). Mean cumulative NDVI (cNDVI) is in direct proportion to averaged NDVI and it is 168 times higher because it covers a period of 168 d in each year. Standard deviations and coefficients of variations are different because in the case of final cNDVI they take into consideration variability of cNDVI between years (not within seasons). The highest relative temporal variability of cNDVI was observed for Denmark (CV = 6.4%), Romania (CV = 5.9%) and Bulgaria (CV = 5.4%), while the lowest temporal variability was observed for France (CV = 2.1%). This means that in the case of France, very similar values of cNDVI were observed across the years of the study (2010–2018).

*3.2. Relationships between NDVI and Grain Yield*

For evaluation of the relationships between MODIS derived NDVI and grain yield of cereals (for all species together and separately for wheat and barley), correlation coefficients were calculated. The calculations were based on the data for 2010–2018 (N = 9) and values of NDVI for the analyses were used for every 8-day period from the end of February to the beginning of August (21 different 8-day periods). Correlation coefficients between NDVI and grain yield of all cereals are presented in Table 3. For different countries, vari-

ous strengths of correlations were observed in different time periods. Very strong positive correlations (correlation coefficients of about 0.7 or greater) at the end of February and in March (for days of the year from 57 to 89) were observed for Latvia, Lithuania, Croatia, Czechia, Germany and Slovakia. This means that the grain yield of cereals in total can be predicted in these countries from very early spring, i.e., about 4 months before the harvest. Strong positive correlations (correlation coefficients about 0.8) were observed a little later, i.e., at the end of March, for Poland, Estonia and Hungary. Strong positive correlations for these countries were observed usually to the end of April and in May and correlations for the first half of June were much weaker. Very strong positive correlations between NDVI and grain yield at the beginning of June were observed in Sweden and Finland, i.e., countries which are located in the north of Europe where harvest of most of cereals is delayed in comparison to other countries included in the study. Another pattern for correlations between NDVI and grain yield was observed in United Kingdom and Ireland where the strongest correlations were observed at the end of May or in the beginning of June. For two countries, very weak or even negative correlations between NDVI and grain yield were observed, i.e., for Belgium and France. This means that yield prediction based on NDVI for these two countries is not possible. Similar correlations as for cereals in total were observed between NDVI with grain yield of wheat and barley (Tables S2 and S3). In the beginning of spring very strong correlations (correlation coefficients about 0.7 or greater) were observed for Czechia, Denmark, Germany, Latvia, Lithuania, Poland and Slovakia.

The correlation coefficients between cumulative NDVI (cNDVI) and grain yield of all cereals are presented in Table 4 and for wheat and barley in Tables S4 and S5. These correlations with cNDVI are much more stable between subsequent dates in comparison to correlations with NDVI. The strongest positive correlations were observed for Croatia, Czechia, Germany, Hungary, Latvia, Lithuania, Poland and Slovakia. For these countries, strong relationships between cNDVI and grain yield of cereals occurred for all dates included in the study. For some countries, the strength of correlation increased from early spring to later dates. Such increase of the correlations was observed for Austria, Denmark, Slovenia and Sweden. For Belgium and France, the correlation coefficients between cNDVI and grain yield were negative along all the seasons. For Bulgaria, Finland, France, Ireland, Romania and United Kingdom the relationships for all studied periods were relatively weak (weaker in comparison to maximal correlations with NDVI).

For most of the countries, the correlations between cNDVI and grain yield of all cereals were similar to correlations with grain yield of wheat and barley. The main exceptions were Croatia where correlations with grain yield of wheat and barley were much weaker in comparison to correlations with grain yield of all cereals. A similar situation was seen in the case of Hungary, but the differences between corresponding coefficients of correlations were smaller.

**Table 3.** The correlation coefficients between NDVI and grain yield of all cereals in 2010–2018. The background color of the cells with the correlation coefficient values indicates the strength of the relationship.

| Days of the Year | Austria | Belgium | Bulgaria | Croatia | Czechia | Denmark | Estonia | Finland | France | Germany | Hungary | Ireland | Latvia | Lithuania | Poland | Romania | Slovakia | Slovenia | Sweden | United Kingdom |
|---|---|---|---|---|---|---|---|---|---|---|---|---|---|---|---|---|---|---|---|---|
| 57–65 | 0.52 | −0.46 | 0.32 | 0.78 | 0.79 | 0.70 | 0.52 | 0.19 | −0.51 | 0.76 | 0.52 | 0.23 | 0.88 | 0.85 | 0.59 | 0.23 | 0.75 | 0.46 | 0.68 | 0.15 |
| 65–73 | 0.54 | −0.49 | 0.55 | 0.79 | 0.77 | 0.68 | 0.57 | 0.06 | −0.50 | 0.67 | 0.73 | 0.27 | 0.90 | 0.82 | 0.58 | 0.44 | 0.84 | 0.35 | 0.26 | 0.06 |
| 73–81 | 0.47 | −0.34 | 0.68 | 0.71 | 0.82 | 0.69 | 0.60 | 0.26 | −0.22 | 0.81 | 0.70 | 0.21 | 0.83 | 0.79 | 0.78 | 0.64 | 0.83 | 0.38 | 0.72 | 0.18 |
| 81–89 | 0.53 | −0.17 | 0.60 | 0.67 | 0.78 | 0.61 | 0.87 | 0.34 | −0.03 | 0.80 | 0.81 | 0.23 | 0.83 | 0.82 | 0.82 | 0.53 | 0.78 | 0.44 | 0.52 | 0.13 |
| 89–97 | 0.55 | −0.25 | 0.50 | 0.70 | 0.78 | 0.66 | 0.73 | 0.39 | −0.17 | 0.54 | 0.78 | 0.31 | 0.79 | 0.75 | 0.58 | 0.49 | 0.66 | 0.45 | 0.49 | 0.32 |
| 97–105 | 0.77 | −0.21 | 0.31 | 0.45 | 0.79 | 0.70 | 0.67 | 0.23 | −0.50 | 0.59 | 0.78 | 0.28 | 0.58 | 0.58 | 0.78 | 0.57 | 0.72 | 0.55 | 0.66 | 0.23 |
| 105–113 | 0.71 | −0.50 | 0.26 | 0.60 | 0.71 | 0.52 | 0.63 | 0.16 | −0.50 | 0.29 | 0.82 | 0.25 | 0.76 | 0.62 | 0.50 | 0.55 | 0.70 | 0.72 | 0.56 | 0.20 |
| 113–121 | 0.59 | −0.66 | 0.16 | 0.64 | 0.76 | 0.40 | 0.56 | 0.26 | −0.19 | 0.30 | 0.86 | 0.51 | 0.58 | 0.61 | 0.53 | 0.44 | 0.80 | 0.69 | 0.28 | 0.49 |
| 121–129 | 0.70 | 0.46 | 0.14 | 0.86 | 0.84 | 0.25 | 0.28 | −0.26 | 0.23 | 0.72 | 0.84 | 0.28 | 0.28 | 0.45 | 0.24 | 0.22 | 0.88 | 0.84 | 0.25 | 0.43 |
| 129–137 | −0.28 | 0.18 | 0.19 | 0.62 | 0.27 | 0.49 | 0.08 | −0.34 | −0.31 | 0.69 | 0.50 | 0.58 | 0.23 | 0.49 | 0.43 | 0.48 | 0.32 | −0.68 | 0.34 | 0.52 |
| 137–145 | −0.01 | −0.29 | −0.09 | 0.35 | 0.38 | 0.49 | −0.25 | −0.56 | −0.02 | 0.63 | 0.27 | 0.74 | −0.21 | 0.17 | −0.32 | 0.13 | 0.30 | 0.07 | 0.10 | 0.59 |
| 145–153 | −0.02 | 0.15 | −0.30 | −0.15 | 0.67 | 0.41 | −0.11 | −0.47 | −0.19 | 0.85 | 0.28 | 0.69 | −0.23 | 0.22 | 0.13 | −0.36 | 0.62 | −0.10 | 0.37 | 0.70 |
| 153–161 | 0.28 | −0.50 | −0.29 | −0.26 | 0.51 | 0.61 | 0.06 | −0.02 | −0.21 | 0.13 | 0.14 | 0.21 | 0.05 | 0.23 | 0.31 | −0.36 | 0.59 | −0.54 | 0.88 | 0.59 |
| 161–169 | 0.48 | −0.01 | −0.09 | −0.03 | 0.23 | 0.77 | 0.25 | 0.48 | −0.52 | 0.07 | 0.16 | −0.05 | 0.25 | 0.18 | 0.49 | −0.21 | 0.52 | 0.14 | 0.77 | −0.03 |
| 169–177 | 0.23 | −0.42 | 0.20 | 0.45 | 0.02 | 0.62 | 0.66 | 0.38 | −0.44 | −0.32 | 0.24 | 0.00 | 0.44 | 0.36 | 0.59 | 0.17 | −0.30 | 0.49 | 0.55 | −0.08 |
| 177–185 | 0.07 | −0.26 | 0.33 | 0.68 | 0.05 | 0.66 | 0.39 | 0.74 | −0.32 | 0.30 | 0.20 | 0.57 | 0.40 | 0.36 | 0.63 | 0.06 | −0.21 | 0.36 | 0.79 | −0.07 |
| 185–193 | −0.04 | 0.16 | 0.50 | 0.85 | −0.14 | 0.55 | 0.59 | 0.66 | 0.00 | 0.27 | 0.33 | 0.63 | 0.56 | 0.25 | 0.51 | 0.51 | −0.30 | 0.13 | 0.80 | −0.15 |
| 193–201 | −0.09 | 0.21 | 0.52 | 0.79 | −0.36 | 0.55 | 0.61 | 0.75 | −0.18 | 0.35 | 0.55 | 0.63 | 0.78 | 0.25 | 0.13 | 0.63 | 0.02 | 0.10 | 0.82 | 0.04 |
| 201–209 | 0.47 | 0.55 | 0.53 | 0.75 | −0.21 | 0.57 | 0.63 | 0.75 | 0.07 | 0.34 | 0.46 | 0.69 | 0.67 | 0.15 | −0.11 | 0.65 | −0.05 | 0.67 | 0.75 | 0.28 |
| 209–217 | 0.48 | 0.28 | 0.49 | 0.69 | −0.08 | 0.56 | 0.41 | 0.55 | −0.09 | 0.43 | 0.39 | 0.68 | 0.39 | −0.12 | −0.24 | 0.61 | 0.00 | 0.95 | 0.64 | 0.01 |
| 217–224 | 0.53 | 0.17 | 0.36 | 0.60 | −0.09 | 0.67 | 0.44 | 0.59 | 0.09 | 0.38 | 0.30 | 0.65 | 0.27 | −0.20 | −0.09 | 0.50 | −0.08 | 0.92 | 0.59 | −0.01 |

Legend: −1.0 −0.9 −0.8 −0.7 −0.6 −0.5 −0.4 −0.3 −0.2 −0.1 0.0 0.1 0.2 0.3 0.4 0.5 0.6 0.7 0.8 0.9 1.0

very strong negative correlation — lack of correlation — very strong positive correlation

**Table 4.** Correlation coefficients between NDVI and grain yield of wheat and barley in 2010–2018. The background color of the cells with the correlation coefficient values indicates the strength of the relationship.

| | −1.0 | −0.9 | −0.8 | −0.7 | −0.6 | −0.5 | −0.4 | −0.3 | −0.2 | −0.1 | 0.0 | 0.1 | 0.2 | 0.3 | 0.4 | 0.5 | 0.6 | 0.7 | 0.8 | 0.9 | 1.0 |
|---|---|---|---|---|---|---|---|---|---|---|---|---|---|---|---|---|---|---|---|---|---|
| | very strong negative correlation | | | | | | | | | | lack of correlation | | | | | very strong positive correlation | | | | | |

| Day of the Year | Austria | Belgium | Bulgaria | Croatia | Czechia | Denmark | Estonia | Finland | France | Germany | Hungary | Ireland | Latvia | Lithuania | Poland | Romania | Slovakia | Slovenia | Sweden | United Kingdom |
|---|---|---|---|---|---|---|---|---|---|---|---|---|---|---|---|---|---|---|---|---|
| **65** | 0.52 | −0.46 | 0.32 | 0.78 | 0.79 | 0.70 | 0.52 | 0.19 | −0.51 | 0.76 | 0.52 | 0.23 | 0.88 | 0.85 | 0.59 | 0.23 | 0.75 | 0.46 | 0.68 | 0.15 |
| **73** | 0.54 | −0.49 | 0.45 | 0.84 | 0.78 | 0.69 | 0.55 | 0.12 | −0.52 | 0.72 | 0.64 | 0.25 | 0.89 | 0.84 | 0.60 | 0.34 | 0.81 | 0.41 | 0.53 | 0.10 |
| **81** | 0.52 | −0.45 | 0.55 | 0.80 | 0.80 | 0.70 | 0.57 | 0.17 | −0.43 | 0.76 | 0.67 | 0.24 | 0.88 | 0.84 | 0.68 | 0.46 | 0.83 | 0.40 | 0.59 | 0.14 |
| **89** | 0.54 | −0.40 | 0.56 | 0.78 | 0.81 | 0.68 | 0.64 | 0.21 | −0.35 | 0.79 | 0.72 | 0.24 | 0.88 | 0.86 | 0.75 | 0.48 | 0.83 | 0.42 | 0.58 | 0.13 |
| **97** | 0.54 | −0.40 | 0.56 | 0.78 | 0.82 | 0.69 | 0.68 | 0.25 | −0.33 | 0.78 | 0.75 | 0.26 | 0.88 | 0.86 | 0.75 | 0.49 | 0.81 | 0.43 | 0.59 | 0.18 |
| **105** | 0.61 | −0.36 | 0.53 | 0.76 | 0.84 | 0.70 | 0.70 | 0.25 | −0.36 | 0.76 | 0.77 | 0.27 | 0.86 | 0.84 | 0.80 | 0.51 | 0.82 | 0.48 | 0.61 | 0.20 |
| **113** | 0.63 | −0.38 | 0.51 | 0.76 | 0.84 | 0.68 | 0.71 | 0.25 | −0.38 | 0.71 | 0.80 | 0.27 | 0.86 | 0.81 | 0.78 | 0.52 | 0.83 | 0.53 | 0.61 | 0.20 |
| **121** | 0.64 | −0.43 | 0.48 | 0.76 | 0.85 | 0.66 | 0.71 | 0.26 | −0.37 | 0.69 | 0.82 | 0.30 | 0.85 | 0.79 | 0.76 | 0.51 | 0.84 | 0.56 | 0.58 | 0.24 |
| **129** | 0.64 | −0.38 | 0.46 | 0.78 | 0.85 | 0.63 | 0.71 | 0.23 | −0.35 | 0.70 | 0.83 | 0.31 | 0.81 | 0.77 | 0.73 | 0.50 | 0.85 | 0.59 | 0.55 | 0.26 |
| **137** | 0.62 | −0.38 | 0.45 | 0.78 | 0.84 | 0.63 | 0.68 | 0.20 | −0.36 | 0.73 | 0.82 | 0.33 | 0.79 | 0.76 | 0.72 | 0.50 | 0.83 | 0.55 | 0.54 | 0.28 |
| **145** | 0.61 | −0.40 | 0.45 | 0.80 | 0.84 | 0.63 | 0.65 | 0.12 | −0.34 | 0.74 | 0.82 | 0.35 | 0.77 | 0.75 | 0.70 | 0.50 | 0.83 | 0.56 | 0.53 | 0.29 |
| **153** | 0.61 | −0.38 | 0.43 | 0.81 | 0.85 | 0.62 | 0.64 | 0.05 | −0.36 | 0.76 | 0.82 | 0.36 | 0.77 | 0.77 | 0.70 | 0.48 | 0.83 | 0.57 | 0.53 | 0.31 |
| **161** | 0.63 | −0.39 | 0.42 | 0.81 | 0.85 | 0.62 | 0.64 | 0.05 | −0.37 | 0.76 | 0.83 | 0.36 | 0.78 | 0.78 | 0.71 | 0.46 | 0.84 | 0.56 | 0.56 | 0.33 |
| **169** | 0.66 | −0.39 | 0.41 | 0.82 | 0.87 | 0.64 | 0.64 | 0.12 | −0.43 | 0.76 | 0.86 | 0.35 | 0.79 | 0.79 | 0.74 | 0.45 | 0.87 | 0.58 | 0.59 | 0.32 |
| **177** | 0.70 | −0.42 | 0.42 | 0.85 | 0.88 | 0.67 | 0.66 | 0.15 | −0.49 | 0.76 | 0.86 | 0.35 | 0.80 | 0.79 | 0.76 | 0.46 | 0.87 | 0.62 | 0.62 | 0.32 |
| **185** | 0.71 | −0.44 | 0.43 | 0.87 | 0.90 | 0.70 | 0.66 | 0.22 | −0.53 | 0.79 | 0.87 | 0.38 | 0.81 | 0.81 | 0.79 | 0.45 | 0.86 | 0.64 | 0.66 | 0.31 |
| **193** | 0.72 | −0.43 | 0.45 | 0.89 | 0.90 | 0.75 | 0.67 | 0.25 | −0.52 | 0.82 | 0.87 | 0.42 | 0.82 | 0.81 | 0.84 | 0.47 | 0.86 | 0.64 | 0.71 | 0.29 |
| **201** | 0.73 | −0.40 | 0.48 | 0.89 | 0.91 | 0.80 | 0.68 | 0.29 | −0.52 | 0.84 | 0.86 | 0.47 | 0.82 | 0.81 | 0.86 | 0.49 | 0.85 | 0.62 | 0.76 | 0.28 |
| **209** | 0.77 | −0.31 | 0.50 | 0.89 | 0.90 | 0.84 | 0.70 | 0.33 | −0.51 | 0.84 | 0.87 | 0.51 | 0.83 | 0.82 | 0.87 | 0.52 | 0.86 | 0.65 | 0.80 | 0.30 |
| **217** | 0.78 | −0.25 | 0.52 | 0.89 | 0.90 | 0.85 | 0.71 | 0.40 | −0.51 | 0.83 | 0.88 | 0.54 | 0.84 | 0.81 | 0.88 | 0.55 | 0.87 | 0.70 | 0.82 | 0.29 |
| **224** | 0.82 | −0.21 | 0.52 | 0.89 | 0.87 | 0.86 | 0.71 | 0.46 | −0.48 | 0.82 | 0.87 | 0.56 | 0.83 | 0.78 | 0.90 | 0.56 | 0.86 | 0.77 | 0.83 | 0.28 |

Linear regression analysis was performed to determine how the increase in NDVI by one unit is related to the increase in grain yield of all cereals, wheat and barley. For such analysis the mean NDVI for the period from 26 February to 7 April (from 57th to 97th day of the year) was used for the analysis as an independent variable. The dependent variables were grain yields for all cereals, wheat and barley. The results are present in graphical form in Figure 2 and in Table S6. The results presented in Figure 2 are only for 8 countries (Croatia, Czechia, Germany, Hungary, Latvia, Lithuania, Poland and Slovakia) for which the relationship between NDVI in early spring and grain yield is strong. The results proved there was a similar relationship for most of the countries and because of this, one common regression function was fitted for all 8 countries. In the case of all cereals, the coefficient of regression is equal to about 16.3 which means that increase of NDVI by 0.1 is related to average increase of grain yield of all cereals by 1.63 t/ha. The relationship is very strong, as evidenced by the high value of the coefficient of determination ($R^2 = 0.719$). For wheat and barley, the relationship is still strong but slightly weaker (i.e., $R^2$ is about 0.61 for both crops). In the case of both species, coefficient of regression has similar value, i.e., about 14, which means that increase of NDVI by 0.1 is related to the average increase of grain yield of wheat and barley by about 1.4 t/ha. The analyses of the regression performed separately for each country allow one to obtain slightly different results, i.e., coefficients of regression for most countries were smaller, i.e., in a range from about 5 to about 14. This means that increase of NDVI by 0.1 is related to increase of grain yield of all cereals as well as grain yield of wheat and barley by 0.5–1.4 t/ha. The strongest relationships were observed for Lithuania, Latvia, Czechia, Germany and Slovakia with regression coefficients in a range from 8 to 13. This means that the increase of NDVI by 0.1 is related to increase of grain yield (for all cereals, as well as for wheat and barley) by 0.8 to 1.3 t/ha.

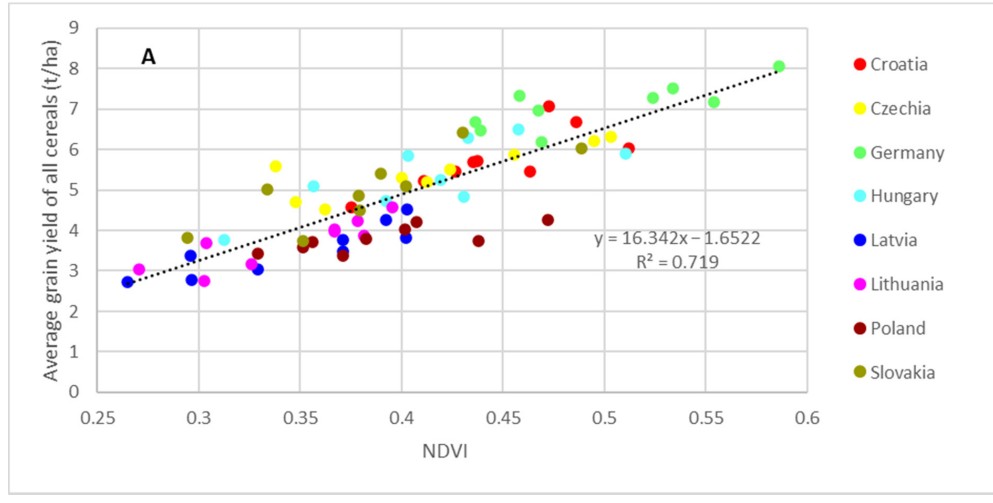

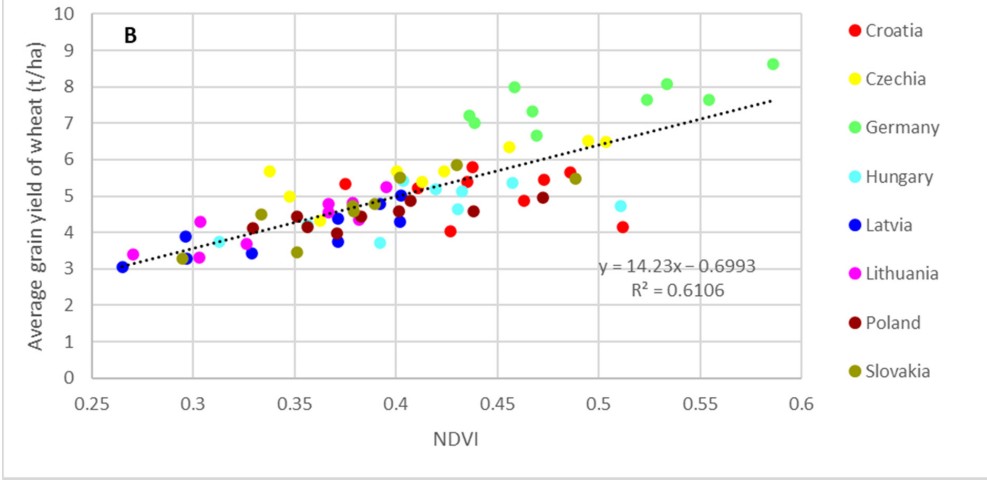

**Figure 2.** *Cont.*

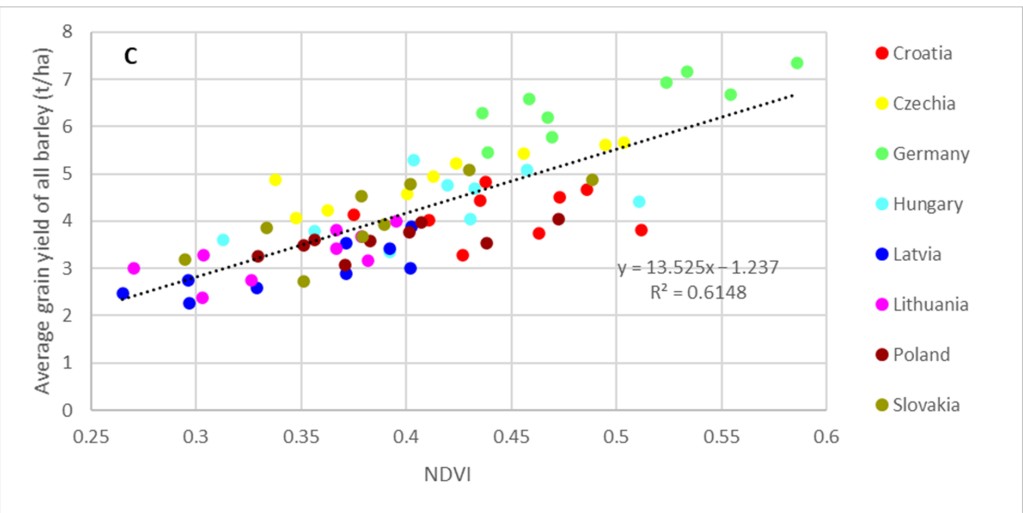

**Figure 2.** The results of linear regression presenting relationships between averaged NDVI (for period of 26 February to 7 April) and grain yield of all cereals (**A**), wheat (**B**) and barley (**C**). Regression equations presented in the plots are based on a dataset consisting of data for 8 countries for which the relationships were strong.

## 4. Discussion

In this study, one of the most promising results is the strong positive relationship between NDVI obtained based on MODIS satellite data in early spring with grain yield of cereals at a country level. Such a relationship was proved for about half of the countries, mostly located in central and eastern Europe. It confirms our previous research conducted at the regional level (based on the data for provinces—NUTS 2) for four countries: Poland, Germany, Czechia and Slovakia [13]. The crop status of the winter cereals, which dominate in the studied countries in early spring, depends on their conditions in late autumn [33,34]. Because of this, it is important for further studies to evaluate the relationships between NDVI in late autumn (e.g., in November) and the grain yield of cereals. One of the problems in such an analysis is the lack of a reliable satellite derived NDVI because of the occurrence of cloud cover during late autumn, which is very common, especially in the northern parts of Europe [35].

The results of regression proved that the increase of NDVI by 0.1 unit is related to the increase of grain yield of cereals by about 1.35–1.65 t/ha based on pooled data. Based on the analyses for individual countries, the slope of regression (coefficient of regression) is usually lower and varies from 0.5 to 1.4 t/ha. These values are slightly smaller than those in the previous study conducted at regional level for central Europe [13]. The different period for which the NDVI was taken for the analyses may be one of the reasons for that. The big differences between coefficients of regression for the relationship between NDVI and grain yield of cereals were observed for different countries of North Africa [36]. The increase of grain yield of cereals related to increase of satellite derived NDVI by 0.1 was, e.g., for Egypt about 0.8 t/ha, while for Morocco it was about 3.7 t/ha. Such a big difference can be explained by different shares of various crop species in these countries. The authors conclude that prediction of the grain yield for a certain area, (e.g., one country) can be based on the simple relationship between NDVI/yield only if the crop area over the observed period is constant. High inter-annual variability of the crop area can weaken and change the relationships [37].

In our study, for some countries located in the northern part of Europe, i.e., Finland and Ireland, the strong positive relationships between NDVI and grain yield were observed in late growth stages which can be related to later development of crops because of lower temperatures. Moreover, for France and Belgium, the relationships between NDVI and grain yield of cereals were very weak or negative along all the seasons, which may be related to small within season as well as between seasons NDVI temporal variability (it is known that in the case of small variability, strength of relationships is weak) [38]. This phenomenon

has been noticed in research, which estimates the relative importance of factors affecting yield and reveals the causes of stagnation in cereal production in France since the mid-1990s. The authors indicate that the climate is generally unfavorable for cereal crops (e.g., high temperatures during grain filling, drought during stem elongation). The greatest influence of the climate was noticed in the areas of intensive cultivation of cereals with high yielding potential. The changes in agrotechnics are also problematic because of: a significant decrease in legume plants in the crop rotation, replaced by rapeseed, and a decrease in nitrogen fertilization were noticed [39,40].

In the Mediterranean climate, the low or even negative correlation coefficients between the durum wheat yield and the NDVI value were also noticed. It was noted that the correlations between grain yield and NDVI were stronger when the correlations between NDVI and growing degree days to heading were weaker [41]. Furthermore, in Sweden, a low or moderate relationship between the yield and NDVI and other vegetation indices was shown, with the emphasis that nonetheless strong associations were identified at different growth stages of crops [42].

## 5. Conclusions

The relationships between NDVI in early spring and grain yield of cereals were very strong for about half of the countries included in our study. This means that the yield prediction for these countries can be done even as early as about 4 months before the harvest. The increase of NDVI in early spring was related to increase of grain yield by about 0.5–1.6 t/ha depending on country and cereal species. The cumulative NDVI (or averaged NDVI directly proportional to it) gives a more stable prediction of grain yield along all of the seasons. For some of the countries (especially France and Belgium), it is not possible to use NDVI for grain yield prediction because of the very weak relationships.

**Supplementary Materials:** The following are available online at https://www.mdpi.com/2073-4395/11/2/340/s1, Table S1: Typical sowing dates and dates of main phenological stages of winter wheat across selected EU countries. Table S2: Correlation coefficients between NDVI and grain yield of wheat. Table S3: Correlation coefficients between NDVI and grain yield of barley. Table S4: Correlation coefficients between cNDVI and grain yield of wheat. Table S5: Correlation coefficients between cNDVI and grain yield of barley. Table S6: Results of linear regression between mean NDVI (for period 26 February to 7 April as independent variable) and grain yield (in t/ha as dependent variable) of all cereals, wheat and barley and the root mean square error (RMSE) for estimation of all cereals, wheat and barley yield based on regression equation.

**Author Contributions:** Conceptualization, D.G. and E.P.; methodology, D.G. and E.P.; formal analysis, E.P. and D.G.; investigation, E.P. and D.G.; data curation, E.P. and D.G.; writing—original draft preparation, E.P. and D.G.; writing—review and editing, E.P.; visualization, E.P. and D.G.; supervision, D.G. All authors have read and agreed to the published version of the manuscript.

**Funding:** This research received no external funding.

**Institutional Review Board Statement:** Not applicable.

**Informed Consent Statement:** Not applicable.

**Data Availability Statement:** Publicly available data were used from the websites: https://glam1.gsfc.nasa.gov/ and http://www.fao.org/faostat/en/ (accessed on 12 January 2021).

**Acknowledgments:** Time series NDVI averaged for countries included in the study was downloaded from GIMMS Global Agricultural Monitoring (GLAM) system (https://glam1.gsfc.nasa.gov/, accessed on 12 January 2021) conducted by USDA and NASA. Grain yields were downloaded from FAOSTAT database http://www.fao.org/faostat/en/ (accessed on 12 January 2021). We acknowledge these institutions, i.e., USDA, NASA and FAO for the possibility of using these datasets for this study.

**Conflicts of Interest:** The authors declare no conflict of interest.

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
