# Peer review of "Relationship between MODIS Derived NDVI and Yield of Cereals for Selected European Countries"

_agronomy, doi:10.3390/agronomy11020340_

Round 1

Reviewer 1 Report

The importance of the study that it evaluates the grain forecast accuracy in wider geographical region: EU. It assesses forcasting possibilities  from 20 countries. The study is well written, with some minor spelling mistakes (e.g. in row 53). Only recommendation is to give more explanations for negative or very weak correlations in certain countries in the Discussion chapter. 

As a summary, I think it is ok after minor modification.

Author Response

The importance of the study that it evaluates the grain forecast accuracy in wider geographical region: EU. It assesses forcasting possibilities  from 20 countries. The study is well written, with some minor spelling mistakes (e.g. in row 53).

Authors: The article was carefully rereading. Some minor spelling mistakes were eliminated.

Only recommendation is to give more explanations for negative or very weak correlations in certain countries in the Discussion chapter. 

Authors: More explanations for negative or very weak correlations in certain countries was added in the Discussion chapter (lines 355-360).

As a summary, I think it is ok after minor modification.

Authors: Thank you very much for all comments. It allowed us to improve quality of the manuscript.

Reviewer 2 Report

The manuscript explores the relationships between NDVI derived from MODIS satellite data and county-level yield of grain yield of all cereals, wheat, and barley across European countries. The results indicate that, for about half of the countries involved in the study, the grain yield based can be predicted four months ahead based on very strong relationships with NDVI in early spring. The manuscript is within the theme of the journal and interesting for potential readers, and thus it could be considered for publications if the following concerns can be solved properly:

- Main contribution and novelty of the work should be presented clearly in the last paragraph of the introduction section.

- Lines 31-34: please give at least one reference regarding yield prediction using statistical models and meteorological data.

- I suggest having information about general crop calendar across countries that can be added in the Section 2.1. I think that information can be used to justify the variation of the relationship between NDVI and grain yield.

- The caption of Tables 3&4 should explain the colours used.

- Please check spelling and typos, for example, lines 29-30.

Author Response

The manuscript explores the relationships between NDVI derived from MODIS satellite data and county-level yield of grain yield of all cereals, wheat, and barley across European countries. The results indicate that, for about half of the countries involved in the study, the grain yield based can be predicted four months ahead based on very strong relationships with NDVI in early spring. The manuscript is within the theme of the journal and interesting for potential readers, and thus it could be considered for publications if the following concerns can be solved properly:

- Main contribution and novelty of the work should be presented clearly in the last paragraph of the introduction section.

Authors: The main contribution and novelty of the work was added in lines 66-70: “This is important because forecasting yields and monitoring the state of agricultural production are key to maintaining Europe's food security. This article is intended to show not only how the yield dependence on NDVI changes in individual European countries, but also to indicate in which countries, based on NDVI, it is not possible to accurately predict yields.”

- Lines 31-34: please give at least one reference regarding yield prediction using statistical models and meteorological data.

Authors: Three various reference was added in line 33. This references refers to various crop models and meteorological indicators in forecasting crop yields.

- I suggest having information about general crop calendar across countries that can be added in the Section 2.1. I think that information can be used to justify the variation of the relationship between NDVI and grain yield.

Authors: The crop calendar across countries was added in Supplementary material (Table S1: Typical sowing dates and dates of main phenological stages of winter wheat across selected UE countries.) (lines 92-98)

- The caption of Tables 3&4 should explain the colours used.

Authors: The explaining of colours in Tables 3&4 was added (lines 269, 293).

- Please check spelling and typos, for example, lines 29-30.

Authors: The article was carefully rereading. Minor spelling mistakes and typos were eliminated.

Reviewer 3 Report

The aim of this study was to evaluate the forecast accuracy of grain yield of cereal crops for selected European countries based on average and cumulative MODIS derived NDVI for different dates in 2010-2018 years. The manuscript is well structured and organized, according to the journal directions (Introduction; Material and Methods; Results, Discussion, Conclusions).

In the reading of the text, I found it quite well written and arranged in the description.

The study topic is surely very much interesting and important for the national economy.

In my opinion, such an analysis is not very precise. The analysis contains many unknowns that may affect the results. It is not known if more forms of spring or winter cereals were studied? They have different ontogenesis, which could have influenced the research results. Is maize included in “cereals all”? In all countries, the measurements were made at the same time. Did the plant's development phase affect the results? It was different in the analyzed countries during the measurements. Could you please, clarify some of these doubts in the paper.  

Table 2 needs improvement. Please explain what is "Average" and "Final cumulative". Which data relates the footnotes under the table. In the text, the authors use the coefficients of variation, but they are not in the table. Please add the coefficients of variation to Table 1. In the Results chapter, Luxembourg is mentioned but not in the tables. Please correct it. NDVI describe to two decimal places as in the table.

In the paper, the authors showed that not every country could use NDVI for forecast yields. The low values of the determination coefficient R2 prove that there is little explanation for the variability. Please include the R2 values in the description of the regression equation.

Author Response

The aim of this study was to evaluate the forecast accuracy of grain yield of cereal crops for selected European countries based on average and cumulative MODIS derived NDVI for different dates in 2010-2018 years. The manuscript is well structured and organized, according to the journal directions (Introduction; Material and Methods; Results, Discussion, Conclusions).

In the reading of the text, I found it quite well written and arranged in the description.

The study topic is surely very much interesting and important for the national economy.

In my opinion, such an analysis is not very precise. The analysis contains many unknowns that may affect the results. It is not known if more forms of spring or winter cereals were studied? They have different ontogenesis, which could have influenced the research results. Is maize included in “cereals all”?

Authors: The data was taken from FAOSTAT website, where FAOSTAT defines cereals as: wheat, rice paddy, barley, maize, popcorn, rye, oats, millets, sorghum, buckwheat, quinoa, fonio, triticale, canary seed, mixed grain, cereals nes. However, in most of the studied countries wheat (especially winter wheat) and barley have large share, as can be seen in Table 1. Other crop species have small or negligible share. The description of cereals was added in section 2.1 in lines 86-90.

In all countries, the measurements were made at the same time. Did the plant's development phase affect the results? It was different in the analyzed countries during the measurements. Could you please, clarify some of these doubts in the paper.  

Authors: The crop calendar for most important crop i.e. winter wheat, across countries was added in Supplementary material (Table S1: Typical sowing dates and dates of main phenological stages of winter wheat across selected UE countries.) (lines 92-98). The differences between countries for winter crops in early spring are not very large because typical growth stage in early spring for winter cereals such as wheat or barley is tillering stage. Bigger differences occur in later growth stages, e.g. flowering stage which can be different by about one month between the studied countries or even within the country the time differences are until 3 weeks.

Table 2 needs improvement. Please explain what is "Average" and "Final cumulative".

Authors: The explanation of “Average” and “Final cumulative” was extended above the Table 2 in lines: 181- 184.

Which data relates the footnotes under the table.

Authors: The explanation of which data relates the footnotes under the table was added in the first row in the Table 2.

In the text, the authors use the coefficients of variation, but they are not in the table. Please add the coefficients of variation to Table 1.

Authors: The coefficients of variation were added in the Table 2.

In the Results chapter, Luxembourg is mentioned but not in the tables. Please correct it.

Authors: The Result chapter was carefully reread and the correction was done.

NDVI describe to two decimal places as in the table.

Authors:  The correction was done.

In the paper, the authors showed that not every country could use NDVI for forecast yields. The low values of the determination coefficient R2 prove that there is little explanation for the variability. Please include the R2 values in the description of the regression equation.

Authors:  Coefficients of determinations are presented in Table S6 for each regression equation. Explanation which presents possible reasons (especially low NDVI in season variability) was added to the manuscript.